# Segregated oceanic crust trapped at the bottom mantle transition zone revealed from ambient noise interferometry

Jikun Feng [1,2✉], Huajian Yao [1,2,3✉], Yi Wang[1,2], Piero Poli[4] & Zhu Mao[1,2]

The recycling of oceanic crust, with distinct isotopic and chemical signature from the pyrolite mantle, plays a critical role in the chemical evolution of the Earth with insights into mantle circulation. However, the role of the mantle transition zone during this recycling remains ambiguous. We here combine the unique resolution reflected body waves (P410P and P660P) retrieved from ambient noise interferometry with mineral physics modeling, to shed new light on transition zone physics. Our joint analysis reveals a generally sharp 660-km discontinuity and the existence of a localized accumulation of oceanic crust at the bottom mantle transition zone just ahead of the stagnant Pacific slab. The basalt accumulation is plausibly derived from the segregation of oceanic crust and depleted mantle of the adjacent stagnant slab. Our findings provide direct evidence of segregated oceanic crust trapped within the mantle transition zone and new insights into imperfect whole mantle circulation.

[1] Laboratory of Seismology and Physics of Earth's Interior, School of Earth and Space Sciences, University of Science and Technology of China, Hefei, China. [2] CAS Center for Excellence in Comparative Planetology, University of Science and Technology of China, Hefei, China. [3] Mengcheng National Geophysical Observatory, University of Science and Technology of China, Mengcheng, China. [4] University Grenoble Alpes, CNRS, ISTerre, Grenoble, France. ✉email: jkfeng@mail.ustc.edu.cn; hjyao@ustc.edu.cn

Recycled oceanic crust plays an important role in generating mantle heterogeneities with critical insights into underlying mantle material circulation[1–3]. However, the behavior of oceanic crust component of slabs during subduction within the mantle transition zone (MTZ) is poorly constrained. The MTZ, bounded by global seismic discontinuities near 410- and 660-km, plays a fundamental role in mantle dynamics and material circulation[2,4–6]. Subducted slabs encounter significant resistance at the 660-km discontinuity and only part of these slabs can directly penetrate into the lower mantle[6,7]. Subducted slabs were proposed to go through composition segregation of the oceanic crust and underlying depleted components at the base of the MTZ due to the density crossover generated by the difference in the depth range of post-spinel and post-garnet transformation, subsequently the separated oceanic crust may be gravitationally trapped at the bottom of the MTZ[2,4,8]. This process is essential for generating chemical and isotopic heterogeneities within the MTZ and critical for our understanding of mantle circulation[2,5,9]. However, this segregation process at the base of the MTZ during slab penetration is challenged because of the limited depth interval of density crossover and high viscosity of cold slabs[10,11]. And the recycling of oceanic crust has long remained a matter of debate[12].

So far indirect geochemical observations[9] and thermochemical predication[8,13] provide the major evidence for inferring recycled oceanic crust within the MTZ. However, these studies are incapable to resolve the current status of the MTZ. Direct high-resolution seismic observation of mineral heterogeneities within the MTZ will put new perspectives on mantle material circulation.

To tackle this challenge and better characterize localized mineral heterogeneities near the stagnant Pacific slab in east Asia, we here combine high-resolution observation of reflected body waves (P410P and P660P) retrieved from ambient noise inter-ferometry (ANI) with mineral physics modeling. ANI provides new opportunities to investigate the Earth's interior structures without event-station geometry limitation. It has been well established that the Green's function between a station pair can be estimated by cross-correlation of continuous ambient noise records[14,15]. In addition to the predominant surface wave signals in the ambient noise cross-correlation functions (NCFs), the recovery of body waves (e.g. P410P and P660P) receives increasing attention and is employed to explore internal structure of the Earth[16–18].

## Results

**Imaging mantle discontinuities from seismic interferometry.** Abundant seismic records make North China Craton (NCC) a perfect natural laboratory to study the interaction between the stagnant Pacific slab and surrounding mantle (Fig. 1). The con-tinuous vertical component waveforms recorded by a temporal dense seismic array and permanent seismographs within NCC were cross-correlated to retrieve the interstation reflected body waves from the MTZ interfaces (Fig. 1c). Retrieved P410P and P660P were adopted to further constrain the spatial variation of MTZ discontinuities and potential mineral heterogeneities in the vicinity of the Pacific subduction zone.

All NCFs were first arranged by their interstation distance, with the result revealing distinctly, both Rayleigh waves and reflected body waves (P410P and P660P) on the NCF section (Fig. 1c and Supplementary Fig. 1). Thanks to the large dataset (74,114 NCFs in total), the recovered reflected phases exhibit high signal-to-noise ratio. The arrival times of reflected body waves in the NCFs have a good agreement with the theoretical traveltime curves (Supplementary Fig. 1). However, the reflected body waves from the 410- and 660-km interfaces interfere with the Rayleigh wave trains within some distance ranges. We remove this interference with a curvelet filtering technique[19,20]. All the NCFs were filtered into two period bands (2–5 s, 5–10 s) and stacked within a series of overlapped distance bins before curvelet

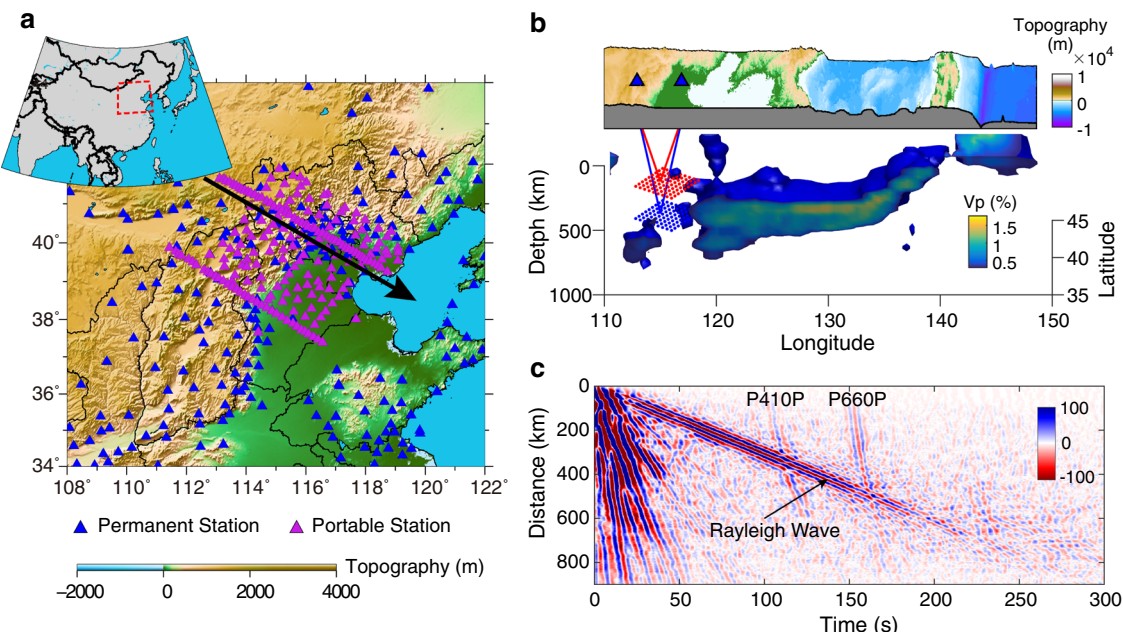

**Fig. 1 Station map, imaging region, and ambient noise cross-correlations. a** Station map with blue and purple triangles representing permanent and portable stations, respectively. Black arrowed line indicates the selected profile shown in Fig. 3. The study region is marked as a red dashed box in the upper left inset. **b** Tomography image and the shape of the high-speed Pacific slab, the shape of slab is calculated from the seismic model from ref. [38]. Blue triangles represent two example stations. Red and blue lines show the raypath geometries of P410P and P660P phases, red and blue dots indicate the reflection points at the 410- and 660-km discontinuities, respectively. **c** Ambient noise cross-correlations filtered to the 2–10 s period band and stacked within a series of overlapped distance bins, where the width of bins is set to 19 km.

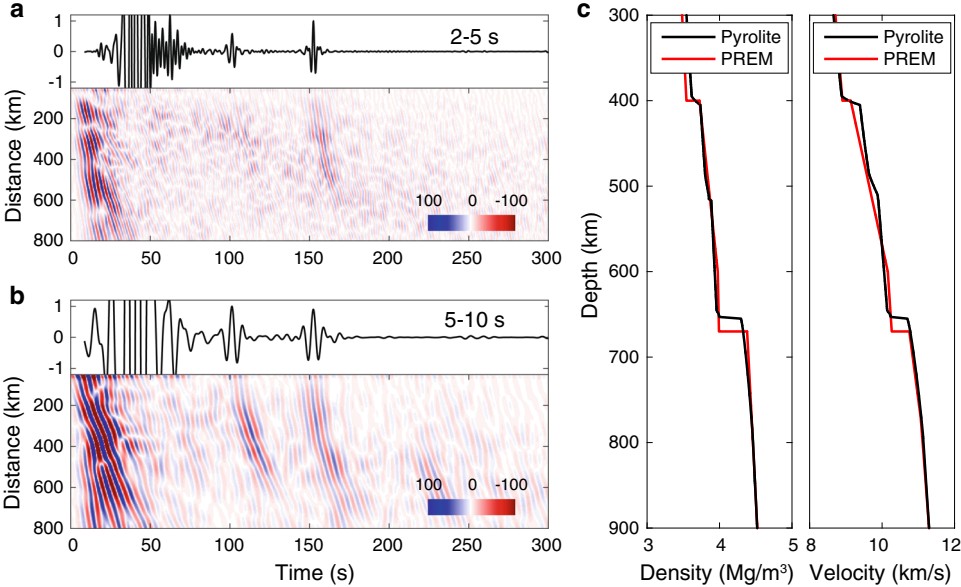

**Fig. 2 Frequency dependence of reflected waveforms and seismic structures. a** Seismogram (top black trace) calculated from the pyrolite model shown in **c** and filtered to 2–5 s and separated reflected body waves filtered to the same period band. All the ambient noise cross-correlations were first regularized by stacking within a series of overlapped 19-km wide distance bins before the seismic phase separation with a curvelet filtering technique. **b** Same as **a** but filtered to 5–10 s. **c** Seismic profiles of the pyrolite model with the Preliminary Reference Earth Model (PREM)[24] overlapped for comparison.

filtering, to investigate discontinuity sharpness and frequency dependence of reflected body waves.

The reflected body waves show different features in two different period bands (Fig. 2a, b). In the longer period band (5–10 s), the body waves reflected at 410- and 660-km interfaces are both clear and exhibit similar amplitudes. By contrast, in the shorter period band (2–5 s), a clear and continuous P660P signal can also be easily identified on the NCF section, even with slightly lower signal-to-noise ratio than that at longer periods, but the P410P phase is very weak or invisible.

Though the contribution from coda waves after large earthquakes may bias the relative amplitude information in the NCFs and introduce spurious branches[21,22], the contribution from coda waves usually predominates the longer periods (>25 s) and is not so significant at shorter periods (<10 s). At shorter periods, the reconstruction of body wave phases is not affected by earthquake coda waves, with relative amplitudes close to the Green's function expected for the actual Earth response[22]. Here, differentiation of NCFs has been calculated for body wave phases to estimate the Green's functions[23]. The excellent lateral coherence of the recovered reflected phases further enhances our confidence in the observations (Fig. 2 and Supplementary Fig. 1), and the lateral variation in relative amplitudes was also tested to be robust (Supplementary Figs. 8–10).

**Regional average structure and mineral composition.** While most 1-D seismic velocity models (such as the Preliminary Reference Earth Model (PREM)[24]) only show a first-order impedence contrast at MTZ discontinuities, more complex structures can exist in the proximity of the MTZ. To evaluate the regional average MTZ structure beneath our study region, and potential mineral composition, thermodynamic calculations were performed to obtain seismic velocity profiles for given mantle compositions[25,26].

We first calculated a seismic velocity profile for pyrolite composition[27] along a 1600-K adiabat (black lines in Fig. 2c). The pyrolite composition, at normal temperature, generates a broad 410-km discontinuity (~10 km) and a very sharp 660-km

discontinuity instead of the two first-order impedance jumps as indicated by PREM. Furthermore, the lower boundary of the MTZ of the pyrolite composition is ~10 km shallower than that of PREM (Fig. 2c). Then the analytical solution was calculated from the obtained seismic velocity profile at an epicenter distance of 200 km with the FK package[28]. The synthetic waveform was also filtered into the two corresponding period bands and normalized within the target time window (80–200 s).

The synthetic waveforms also show obvious frequency dependence in line with our seismic observations (Fig. 2a, b), which is not expected for both sharp 410- and 660-km discontinuities (Supplementary Fig. 7a, b). One plausible conclusion is that on average a sharper 660-km discontinuity exists beneath this region in line with mineralogical prediction[29,30], compared with the 410-km discontinuity. A consistent conclusion was also obtained through fitting the waveform of P410P and P660P in NCFs[17], which agrees with Lawrence and Shearer[31] who suggest a ~3 times thicker 410-km discontinuity relative to the 660-km discontinuity. However, both sharp 410- and 660-km discontinuities were also reported beneath the Indian Ocean[32].

**Lateral variations in reflected waveforms and composition.** Though the average MTZ discontinuity structure beneath the study region seems to have a good agreement with the pyrolite composition, the local structure may be much more complex, due to subduction-induced lateral temperature and/or chemical heterogeneity[33]. Here the phase-weighted stacking method for common reflection points was adopted to explore the detailed structure and potential mineral heterogeneities[16].

Different from the previous time-domain correction and stacking[16], in this study, the time-domain series were directly converted to depth-domain based on the ak135-f model[34], to account for different raypath geometries before common reflection point stacking. Then the depth-domain NCFs were stacked according to their reflection points within each stacking bin following our previous work[16]. The bins were defined with a circle with a radius of 60 km and the bin centers are shown as circles in Supplementary Fig. 4. The stacked waveforms along one

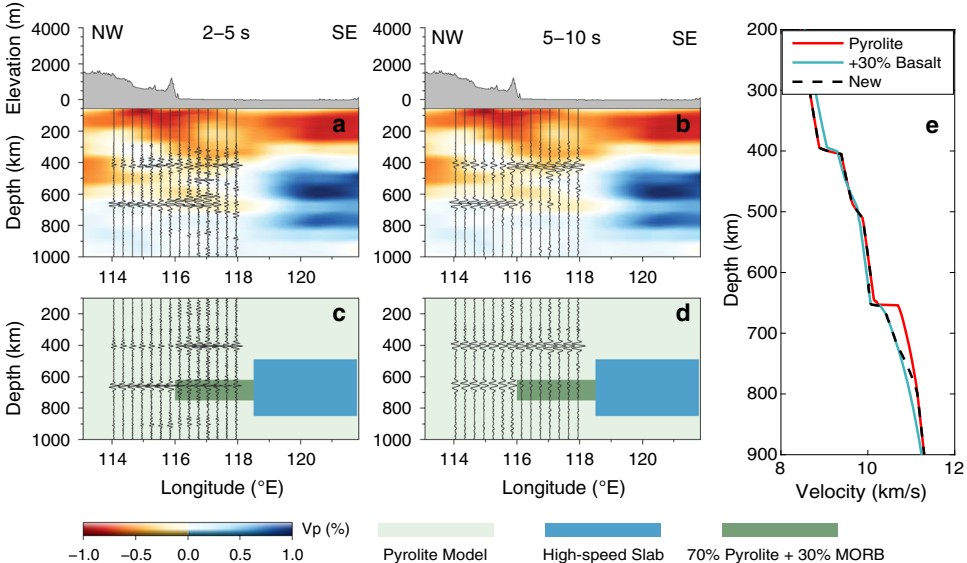

**Fig. 3 Reflected waveforms and the potential mineralogical model. a** The retrieved waveforms reflected at the mantle transition zone discontinuities filtered to 2–5 s and converted to depth-domain. The background color shows the P-wave speed perturbation from ref. [38]. **b** Same as **a** but filtered to 5–10 s. **c** Synthetic waveforms of the reflected phases filtered to 2–5 s and converted to depth-domain. The background color denotes the bulk mineral composition. **d** Same as **c** but filtered to 5–10 s. **e** The P-wave speed profiles for the pyrolite composition (red line), the mixture of 70% pyrolite and 30% Mid-ocean-ridge basalts (MORB) (sky-blue line), and the new model of pyrolite composition with 30% MORB enrichment from 620 to 750 km and fading to normal pyrolite composition from 750 to 800 km (dash black line).

selected profile (black arrowed line in Fig. 1a) were filtered and shown in Fig. 3a, b. High-quality body waves reflected at the MTZ discontinuities can be identified within both period bands. The foremost feature of the stacked profile is the clear amplitude contrast in the waveforms reflected at the 660-km interface from northwest to southeast.

In the northwestern part, our results exhibit stronger P660P phase and weaker P410P phase in the 2–5 s shorter period band and comparable amplitudes of P660P and P410P phases in the 5–10 s longer period band, implying a much sharper 660-km discontinuity. This feature is consistent with the synthetic waveforms calculated from the pyrolite composition. Moreover, the reflected waveforms are simple and show great lateral coherence. Hence, in the northwestern part, the MTZ seems simple and compositionally undisturbed. In contrast, in the southeastern part, the stacked waveforms exhibit distinct characteristics: much more complicated P660P and clear P410P waveforms in the shorter period band, but weak-to-invisible P660P and clear P410P phases in the longer period band near the stagnant Pacific slab. These features of reflected body waves in the southeastern part are very different from the synthetic waveforms from the pyrolite composition, which implies the existence of mineral and/or temperature disturbance.

Even higher temperature can generate a more gradual seismic speed gradient across the lower boundary of the MTZ, resulting in weak P660P phase (Supplementary Figs. 11 and 12)[25]. However, normal to slightly lower temperature seems more reasonable near the subducted slab, which signifies sharp discontinuity and strong P660P phase. Defocusing effect owing to topography variation also seems untenable as no obvious traveltime changes were observed except for more complex waveforms within the shorter period band (Fig. 3a). Hence the abnormal structure likely results from chemical heterogeneity derived from the adjacent trapped slab in the MTZ.

Subducted slabs are composed of oceanic crust (Mid-ocean-ridge basalts, MORB) and underlying depleted mantle (harzburgite and peridotite). In order to evaluate the influence of chemical

heterogeneity from the trapped slab, pyrolite was mixed with MORB or harzburgite in different proportions to calculate corresponding seismic velocity profiles (Fig. 4). Enrichment of harzburgite at the base of the MTZ generates larger impedance contrast across the 660-km interface and stronger reflection. By contrast, enrichment of MORB across the 660-km interface generates a more complicated discontinuity architecture: a small sharp increase of wave speed around 660 km followed by a gradual increase of wave speed to greater depth.

To interpret our observations, a simple mineralogical model was constructed along the profile: background composition of mantle average pyrolite with 30% MORB[35] enrichment at the base of the MTZ just ahead of the high-speed slab as shown in Fig. 3c–e. As our study region is outside the cold slab and no significant high wave speed anomaly resolved beneath the seismic array, a normal adiabat (1600 K) is assumed. At each location, synthetic waveform was calculated for epicenter distance at 100 km. The spectral amplitude of the synthetic seismogram was subsequently normalized and multiplied by the corresponding spectral amplitude of real NCF waveform data[17]. The reflected body waves also were normalized according to our target time window and only the relative amplitude information was retained.

Overall, the synthetic waveforms show an excellent first-order agreement with our observation (Fig. 3c, d). Even the 660-km discontinuity is much broader due to the enrichment of MORB and associated with week-to-invisible P660P at longer periods (5–10 s), the small sudden increase of wave speed ~660 km can also cause obviously P wave reflection at shorter periods (2–5 s) in the southeastern part of the profile. But the observed waveforms at the shorter periods exhibit a remarkable complexity, compared with that from the simple mineralogical model. As the upper boundary of MORB assemblage may also generate reflected signals, the complicated reflected waveforms (Fig. 3a) may imply significant roughness of the accumulated oceanic crust. The accumulated oceanic crust assemblage also exhibits remarkable lateral irregularity, and its scale looks >2° (Supplementary Fig. 5).

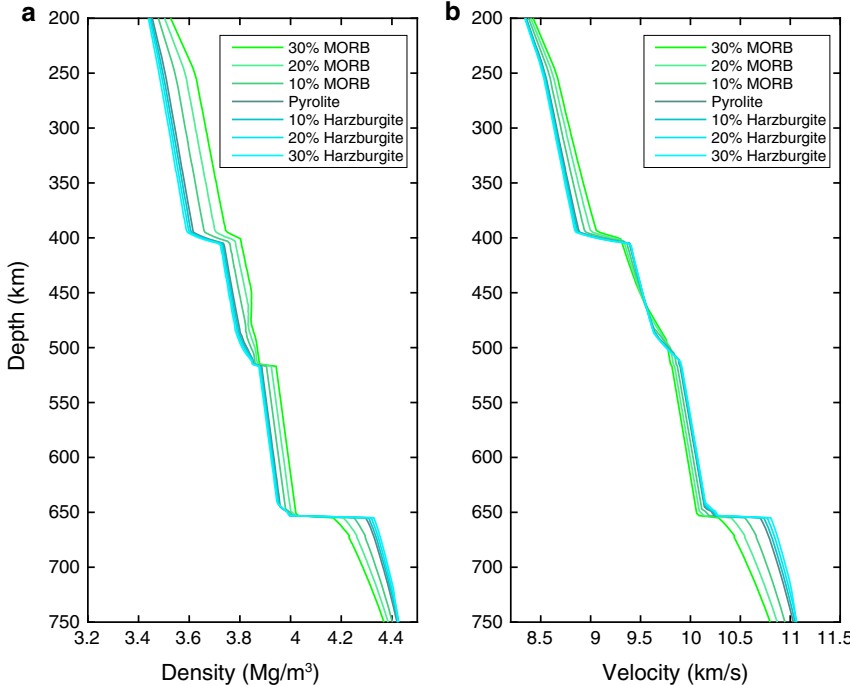

**Fig. 4 Synthetic density and P-wave speed profiles for different compositions. a** Density profiles for the mixtures of *x* harzburgite or Mid-ocean-ridge basalts (MORB) with (1-*x*) pyrolite along a 1600-K adiabat. **b** P-wave speed profiles for the mixtures.

The body waves observed in the northwestern part of the profile are well consistent with that from the simple pyrolite composition.

## Discussion

As inferred from our result, the regional average mantle composition is similar to the pyrolite composition and a localized MORB accumulation exists just ahead of the stagnant Pacific slab. It seems that the compositional contamination from the stagnant Pacific slab is currently localized near its front and other regions are currently compositionally undisturbed.

Though garnet-enriched MTZs have been previously reported by both seismic[5,36] and thermochemical studies[8,13], here new insights on the origin and formation mechanism of garnet-enriched heterogeneities were provided by a high-resolution ANI method. Slab subduction is one of the most important ways for mantle material circulation and a potential source for thermal and chemical heterogeneities[33]. Our observation indicates that at least part of the separated oceanic crust from stagnant slabs is trapped at the bottom of MTZ.

Mineral physics data indicates that MORB (and its high-pressure forms) has a higher density than normal pyrolitic mantle in the upper mantle and the transition zone but a lower density at the bottom transition zone due to the difference in their phase-transition depths[11,37]. Consequently, the separated oceanic crust may be gravitationally trapped at the bottom of the MTZ. The material-filtering effect of the 660-km interface may play a crucial role in the chemical evolution of our planet[2,8].

The fate of oceanic crust of slabs may depend on the subduction style (Fig. 5). Different subducted slab behaviors have been systematically surveyed by high-resolution seismic tomography studies[6,7,38]. Part of the subducted slabs directly penetrates through the MTZ and plunges steeply in the lower mantle. For these slabs, the separation of oceanic crust and underlying depleted components within the MTZ is believed to be difficult due to high viscosity (cold) and fast penetration[8,10]. Such slabs

may undergo composition segregation above the core-mantle boundary and be accumulated at the bottom of the mantle or entrained by upwelling plumes[8,39,40]. However, other subducted slabs, such as the Pacific slab beneath east Asia, were impeded by a barrier around the 660-km depth and deflected horizontally. After a long stagnation within the MTZ, these distorted slabs were warmed up (low viscosity) and may experience composition segregation owing to the distinct contrast in density and rheology of oceanic crust and underlying mantle of subducted slabs[8,41]. Small scale folding is also proposed to contribute to the segregation within the MTZ[42]. Our seismic observation is in favor of such a scenario (Fig. 5). In addition, the stagnant slabs seem just transient features and residual oceanic mantle lithosphere may finally further sink into the lower mantle. The high-speed anomaly beneath the 660-km interface visible on the seismic tomography result (Figs. 1b, 3a, b, and Supplementary Fig. 13) probably denotes the separated oceanic mantle lithosphere.

Our seismic observation and mineralogical modeling signify the accumulation of oceanic crust (MORB) ahead of the stagnant Pacific slab and further the segregation of oceanic crust from underlying melt-depleted components. This clue is of fundamental importance for understanding the evolution and final fate of the oceanic slabs trapped within the MTZ and material circulation process. The chemically filtering effect of the 660-km discontinuity on mantle material circulation is in favor of partially blocked whole mantle convection. The bottom of the MTZ may provide a secondary reservoir for recycled material with "enriched" isotopic signature in addition to the core-mantle boundary[9,43].

Methodologically, this paper adopted a newly developed method[16] sensitive to fine P-wave speed structures at MTZ discontinuities with a much higher resolution than PP precursor waves[44,45]. Although receiver functions (RFs) can also provide resolution on the MTZ discontinuities, they are most sensitive to S-wave speed structures. This ANI technique is an essential complement to traditional seismic tools. Though no significant changes in P410s and P660s amplitude are reported by RF studies

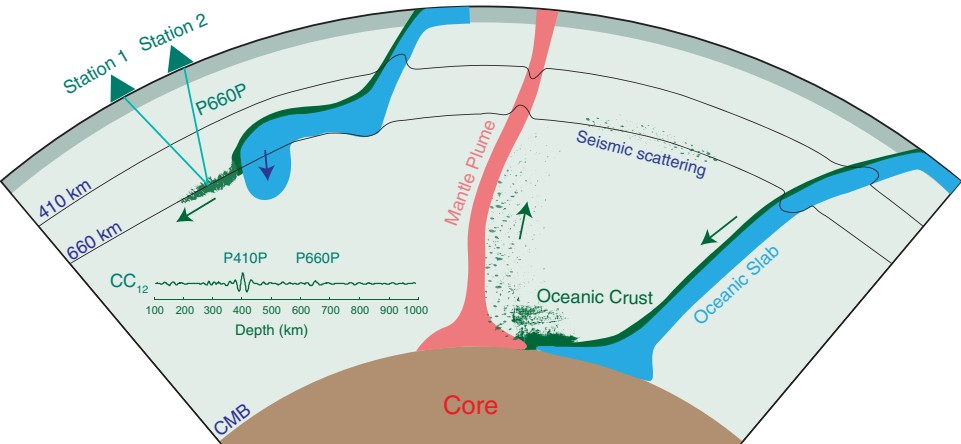

**Fig. 5 Schematic diagram for the evolution of oceanic crust in deep mantle.** Part of the subducted slabs plunges deeply into the lower mantle and undergoes composition segregation above the core-mantle boundary (CMB). The separated oceanic crust component may be entrained by hot plumes. Other subducted slabs encounter insurmountable resistance at the 660-km interface and are trapped within the mantle transition zone (MTZ). These slabs may experience composition segregation within the MTZ and chemical filtering. Accumulation of oceanic crust at the base of the MTZ can explain observed seismic scattering[5] and even weak P660P phase ahead of ancient slabs inferred in this study. The separated depleted components of the slab may finally sink into the lower mantle. $CC_{12}$: cross-correlation of continuous seismic ambient noise records from station 1 and station 2 after conversion to the depth domain.

at the same region, apparently depressed 660-km interface and thicker MTZ can be identified beneath the North China Basin in the newest RF study[46]. The depressed and broadened P660s beneath the North China Basin (Supporting Information for Sun et al.[46]) may be alternatively interpreted as a gradual gradient beneath 660-km depth caused by trapped MORB composition, which is compatible with our model. Therefore converted (RFs) and reflected (P410P and P660P) phases can be further jointly analyzed for better imaging the complex interface structures of the Earth in the future.

## Methods

**Ambient noise cross-correlation calculation.** All continuous ambient noise waveform records were systematically preprocessed and subsequently correlated following Bensen et al.[47]. To calculate the ambient noise cross-correlation functions (NCFs) between station pairs, all the available continuous waveform data were first cut into daily segments. Then the instrument response, data mean, and trend were removed from the daily records followed by spectral whitening and temporal normalization. Here, the running absolute average method was applied in the time domain to suppress earthquake-related nonstationary signals within four period bands (2–5 s, 5–10 s, 10–20 s, and 20–50 s). Lastly, the daily waveform segments were cross-correlated and finally linearly stacked to yield the final NCFs. The symmetric NCFs were also folded at time zero to further enhance coherent signals. As described, all available station pairs were correlated, and a total of 74,114 NCFs were employed in this study. All folded NCFs were filtered to two period bands (2–5 s and 5–10 s) and stacked within a series of interstation distance bins (Supplementary Fig. 1a, b), where the bin width is set to 19 km. Supplementary Fig. 1c exhibits the number of NCFs with respect to interstation distance.

Coherent reflected body waves can be observed within both shorter (2–5 s) and longer (5–10 s) period bands, aligned well along the theoretical traveltime curves of P410P and P660P. However, the reflected body waves within the shorter period band exhibit much poorer signal-to-noise ratio relative to those in the longer period band. That is probably why high-frequency reflected body waves were nearly invisible in the NCFs section in our previous study[16], in which the total number of used NCFs is about one-fourth of those adopted in this study.

It should be noted that the absolute amplitudes of different phases have been lost after time-domain normalization. So, we focus solely on the remarkable relative amplitude features that seem very stable and reliable for different sub-datasets (Supplementary Figs. 8–10).

**Separation of the reflected body waves and Rayleigh waves.** As shown in Fig. 1c, the predominant Rayleigh waves interfere with the weak reflected body waves from 410- and 660-km interfaces in time-space domain. It is difficult to separate these overlapped signals in time-space domain. A curvelet transform technique was implemented to separate these signals in curvelet domain according to the differences in their slopes[20].

Curvelet, consisting of a series of localized "fat" plane waves, can be used to sparsely represent wavefields[19,48]. When transformed to curvelet coefficient domain, the wavefield is expressed as the suppression of curvelets characterized by different scales and directions. The interfering phases in time-space domain can be easily separated from one another in directions. The scale of curvelet corresponds to the frequency content of the data. The code used here and more numerical implementations of curvelet filtering are available at http://www.curvelet.org (last accessed 31 October 2020).

Though the Rayleigh waves interfere with the reflected phases in time-space domain, the slopes of the reflected waves are much steeper than those of the Rayleigh waves. These interfering phases can be easily separated by extracting the corresponding curvelet coefficients. Then the corresponding curvelet coefficients were transformed back to time-space domain to recover the separated waveforms (Supplementary Fig. 2).

**Phase-weighted stacking method for common reflection points.** In this study, a common reflection point stacking method and phase-weighted stacking technique[49] are combined to enhance signal-to-noise ratio and lateral resolution[16]. Before common reflection point stacking, all the NCFs were converted to depth-domain based on a 1-D model[34] to account for different raypath geometries.

The stacking bins for common reflection points are defined within a circle with a radius of 60 km. Despite curvelet filtering technique can be adopted to reduce the interference effect from predominant Rayleigh waves, this method often does not work well within small bins due to the limited amount of data. Here, only the station pairs with distance <200 km were employed to reduce the interference. As the distance between permanent stations is usually too large, only the NCFs from the transportable dense array stations were utilized for further stacking. For each stacking bin, the reflection points of NCFs at the 410- and 660-km interfaces were calculated and determined if locating within the stacking bin. All the NCFs whose reflection points located within the stacking bin were selected and stacked with both linear and phase-weighted stacking methods in depth-domain. The centers of the stacking bins and the number of stacked NCFs within each bin are shown in Supplementary Fig. 4. The blue arrowed lines in Supplementary Fig. 4 indicate three selected profiles (the profile shown in Fig. 3 is BB' profile). The data density is high along three selected profiles and becomes lower towards south. Only the stacked traces along the selected profiles are shown in Supplementary Fig. 5. Because they are superimposed traces of more waveforms, they show much better stability and reliability. The P660P weakening can be easily identified on all three selected profiles.

Due to the limited data coverage and poor signal-to-noise ratio in the south part, it is difficult to exactly outline the range of the P660P weakening zone. Apparently, the shape of this P660P weakening zone is irregular, and its scale seems at least 2° as inferred from Supplementary Fig. 5. The irregular distribution of Mid-ocean-ridge basalts (MORB) is plausibly due to the complex shape of slab front and 3-D mantle structures.

**Mineral physical modeling and waveform calculation.** In this study, the seismic velocity profiles for several mantle compositions (the pyrolite composition and the mixtures of pyrolite with MORB or harzburgite in different proportions) were

calculated following the procedures outlined by Weidner and Wang[25] and Wang et al.[26]. The composition of pyrolite is taken from Workman and Hart[27] and MORB and harzburgite are taken from Baker and Beckett[35]. The phase diagram of the CMAS system reported by Gasparik[50] was adopted in this calculation. For simplicity, pyrolite, MORB, and harzburgite are used to denote specific compositions and their high-pressure forms throughout the mantle depth range.

We first calculated the velocity profiles for the pyrolite composition along a range of mantle adiabats (Supplementary Fig. 11). Then we calculated the velocity profiles for different mineral compositions (mixtures of pyrolite with MORB or harzburgite in different proportions, Fig. 4) along a 1600-K adiabat. In order to evaluate the influence of temperature on reflected waveforms, for each seismic model, the synthetic waveform was calculated with the FK package[28] and the body waves reflected at the MTZ discontinuities were analyzed (Supplementary Fig. 12). Here vertical single force was used for source located at a shallower depth (5 km, the source cannot be set at the free surface for the FK package) and the receiver was placed 100 km away at the free surface. The target phases were further filtered (to two period bands: 2–5 s and 5–10 s) and normalized within our objective time window (80–200 s) as shown in Supplementary Fig. 12.

## Data availability

Waveform data of permanent stations for this study are collected through Data Management Centre of China National Seismic Network at Institute of Geophysics, China Earthquake Administration (https://doi.org/10.11998/SeisDmc/SN, http://www.seisdmc.ac.cn). Waveform data of portable stations are archived at China Seismic Array Data Management Center at Institute of Geophysics, China Earthquake Administration (https://doi.org/10.12001/ChinArray.Data, http://www.chinarraydmc.cn). Restrictions apply to the availability of the raw waveform data; request for the original data needs to be sent directly to the management centers. Ambient noise cross-correlations generated and analyzed during the current study are available in the Mendeley Data repository[51].

## Code availability

The codes to perform ambient noise analysis are available from the corresponding author upon reasonable request. Most figures were generated with Generic Mapping Tools (GMT) (https://www.generic-mapping-tools.org).

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

## Acknowledgements

Jikun Feng is grateful to Chunquan Yu for his help on curvelet filtering and Ling Chen and Wenbo Wu for useful discussions. This research is funded by Youth Fund of National Natural Science Foundation of China (41904048), the Strategic Priority Research Program of Chinese Academy of Sciences (Grant No. XDB 41000000), and National Natural Science Foundation of China (41574034).

## Author contributions

H.Y. conceived and supervised the study. J.F. processed the seismic data and produced figures. Y.W. performed mineral physics modeling. P.P. contributed to ambient noise data analysis. Z.M. contributed to mineralogical discussion and final conclusion. The manuscript was drafted by J.F. and edited by all authors.

## Competing interests

The authors declare no competing interests.
