## [Peer Review File · Nature Communications]

REVIEWER COMMENTS

Reviewer #1 (Remarks to the Author):

The fate of subducted oceanic crust is essential for understanding styles of mantle convection, with accumulated oceanic above 660km interface favoring layered convection and with penetration of oceanic crust into lower mantle favoring whole mantle convection. The manuscript by Feng et al provides very valuable evidences of oceanic crust retained above 66km interface with the method of ambient noise interferometry (ANI). With ANI, they retrieved clear P410P and P660P signals, and observed frequency-dependent amplitude ratio of P660P/P410P as well as lateral variation of the amplitude ratio. Based upon mineralogical modelings and seismic waveform modeling, the authors provide convincing arguments for presence of MORB in the bottom part of mantle transition zone. I would recommend the manuscript to be accepted, after some revisions. Here are some comments:

- 1, Fresnel's zone of P410P and P660P for the two frequency bands. As the authors have observed lateral variation of P660P/P410P in figure S6 and S7, it would helpful to show how rapid variation the dataset might resolve.

- 2, line 209-212. High frequency P'P' precursors (PKP PKP) are important for studies of 410 and 660 interfaces (Day and Deuss,2013,GJI). As for P wave receiver functions, its sensitivity on Vs actually is also important for constraining nature of 410/660km. The authors are encouraged to combine reflected P waves (this study) and previous P410s,P660s datasets to provide tighter constraint.

- 3, line 150. The distance of 100km is adopted here. Why this particular distance?

- 4, Supplementary figures comparing/contrasting P410P,P660P from ANI for the study region in this study and other regions could be helpful. The paper Poli et al (2012) has already some observations of the reflected P waves from mantle interfaces.

S.N.

Reviewer #2 (Remarks to the Author):

The manuscript presents P wave reflection analysis of mantle transition velocity discontinuities to infer variations in mantle composition related to subducted slab remnants beneath eastern China. The method of obtaining small offset P wave reflections relies on ambient noise interferometry and is still relatively new but has been validated in multiple studies. Generally, the recovery of reflections from 410 and 660 is impressive, as is the large scale spatial variability in the relative amplitudes of the two reflections in multiple frequency bands. The transition from a simple sharp 660 far away from slab remnants to a complicated and/or absent 660 reflection closer to slab remnants appears reliable based on the recovery of 410 reflections and the merit of the processing methods. This spatial transition is the key result for the authors' interpretations. The main conclusion is aligned with many studies indicating that the 660 is a leaky boundary in mantle convection that is largely driven by subduction of slabs with two major compositionally distinct components. It's not surprising but this study offers an impressive new angle of evidence that bears on long-standing problems in understanding mantle dynamics and compositional evolution. The text is generally clear and the figure quality is good. I recommend publication after an opportunity for minor revisions.

I'd note that the data used are not publicly available and hence the study is not reproducible by other researchers in the near future, nor is there a set embargo time window after which the data will become open access. I'm not sure if reproducibility is required by Nature Communications and my evaluation above is only based on the scientific merit of the manuscript.

Reviewer #3 (Remarks to the Author):

This is an interesting paper that images mantle transition-zone discontinuities in part of eastern China just west of the subducting Pacific slab and finds evidence for changes in discontinuity properties that the authors argue can be explained as differences in composition related to subduction of oceanic crust. The seismic results are based on ambient noise cross-correlation, a relatively novel approach to studying mantle discontinuities that may have some advantages over more traditional methods. I think the results are potentially important enough to publish in Nature Communications, but I do have some concerns.

(1) Does the noise cross-correlation method accurately retrieve the station-to-station Green's function, specifically the P410P and P660P phases of interest? If the distribution of noise sources is not uniform, then biases and artifacts could be introduced and the assumed bounce-point locations for P410P and P660P might not be correct. Thus, it is important test to evaluate how evenly distributed the incoming noise signals are. Here is one way to perform such a test:

Redo Fig. S1 for two independent subsets of the data, based on the azimuth of the station pair used for the cross-correlation: 0–90 degrees and 90–180 degrees (the division is not for 0–360, as 180 deg. folding of the results is performed (lines 223– 224)). If the images of P410P and P660P don't agree, this suggests that the noise may have some dominant back-azimuths and more work will be needed to make sure the results in the paper are not biased by this non-uniformity. Note: Because of the dense station lines at about 120 deg. azimuth, the 90–180 azimuth bin will have many more cross-correlations, so it might be necessary to perform the test with a more uniform station distribution, i.e., discard data from the stations along these lines, so that the images have roughly equal amounts of data.

(2) It would be good to plot the theoretical P410P and P660P travel-time curves on top of Fig. S1 to make sure they align exactly with the observations.

(3) What causes the reduction in amplitude in P410P and P660P near zero offset? I would not have expected the reflection coefficient to change very much with distance at the relatively steep incidence angles between 0 and 200 km offset. The fold drops near zero offset, but this should increase the noise, not dampen the signal, right?

(4) Given the observed amplitude reduction near zero offset, it makes me nervous that the 200 km cutoff used for the common-reflection-point (CRP) stacking is at a range where the amplitudes are changing. To make sure there is nothing systematically varying with station location that might be biasing the CRP stacks, it would be good to see Fig. S1 repeated using just the data used for the profile shown in Fig. 3, and then plotted separately using just the data for the reflection points in the profile east and west, respectively, of 116.5 longitude (which approximately separates the main changes that are observed). The difference in the P410P and P660P amplitude and character east and west of this point in the profiles is perhaps the main observational result in the paper, so it's important to verify its robustness and make sure it does not have some other possible origin unrelated to the actual mantle discontinuity properties.

(5) The text near line 102 should reference and discuss the Poli et al. 2012 paper, which found a thicker 410- than 660-discontinuity based on P410P and P660P ambient noise cross-correlations, a result more directly relevant to this paper than the other references cited.

(6) There have been several seismic receiver function (RF) studies of mantle discontinuities in the same region (e.g., Chen and Ai, JGR, 2009; Zhang et al., Tectonophysics, 2016; just what I found quickly---there are likely more). Near the end of the paper (~line 210), the authors mention that their method differs from RF studies because it is sensitive to P velocity changes rather than S velocity changes. Nonetheless, one would expect P and S velocity changes across the 410 and 660 to be correlated to some extent and to occur at similar depths. Thus, to put the current paper into better context with previous work, it would be good to include some mention of what the RF studies have found. That is, do they see changes in 410 or 660 amplitude, topography, and/or sharpness that agree or disagree with the topside P reflection data? If there is agreement, this would help support the author's model. If there is disagreement, some discussion and perhaps arguments as to why the new results are better would help the reader in sorting out what is going on.

Note that I am primarily an observational seismologist, so I have focused on those aspects of the paper, and hope that other reviewers will be able to assess the mineral physics modeling.

Reviewer Comments

Reviewer #1 (Remarks to the Author):

The fate of subducted oceanic crust is essential for understanding styles of mantle convection, with accumulated oceanic above 660km interface favoring layered convection and with penetration of oceanic crust into lower mantle favoring whole mantle convection. The manuscript by Feng et al provides very valuable evidences of oceanic crust retained above 66km interface with the method of ambient noise interferometry (ANI). With ANI, they retrieved clear P410P and P660P signals, and observed frequency-dependent amplitude ratio of P660P/P410P as well as lateral variation of the amplitude ratio. Based upon mineralogical modelings and seismic waveform modeling, the authors provide convincing arguments for presence of MORB in the bottom part of mantle transition zone. I would recommend the manuscript to be accepted, after some revisions. Here are some comments:

We appreciate your effort to review our manuscript and your positive feedback. We have tried to revise the manuscript according to your concerns and suggestions. Here below we address the concerns you raised point by point.

1, Fresnel's zone of P410P and P660P for the two frequency bands. As the authors have observed lateral variation of P660P/P410P in figure S6 and S7, it would helpful to show how rapid variation the dataset might resolve.

Thanks for your suggestion. Fresnel's zone is a good conventional tool to evaluate the lateral resolution of target phases. Following your suggestion, we calculate the Fresnel's zone of P_{410P} and P_{660P} for two frequency bands (Fig. R1). The Fresnel zone is defined by a $\pm T/2$ contour, where T is the dominant period (3.5 s and 7.5 s for 2-5 s and 5-10 s period bands, respectively).

To better take into account the special structure discussed in the manuscript, here we design a more targeted test to intuitively evaluate the resolution of the P_{660P} phase to local gradual 660-km discontinuity, which is critical for our discussion in the main text. The designed 2-D models are very simple, which are derived from 1-D iasp91 model with both sharp 410- and 660-km discontinuities. To avoid shallow reverberations within the crust, the velocity structure from free surface to the Moho increases linearly. Local gradual 660-km discontinuity (a linear increase from 660 km to 720 km) is added to test the resolution of the P_{660P} (Fig. R2). Fig. R3 shows the synthetic waveforms calculated with the SPECfEM2D package (Tromp et al., 2008) from three models shown in Fig. R2. Vertical force was set at 300 km on the free surface as source. No apparent frequency-dependence is observed for the body waves reflected from the sharp impedance contrasts at 410 and 660 km (Fig. R3a, b). A 200-km wide gradual 660-km discontinuity can result in remarkably weak P_{660P} phase within two period bands (Fig. R3c, d). On the other hand, a 100-km wide gradual 660-km discontinuity can only generate very weak P_{660P} phase within the shorter period band (2-5 s) (Fig. R3e, f). It can be concluded from this test that a 200-km wide gradual 660-km discontinuity can be resolved by the P_{660P} phase within the longer period band (5-10 s) and a 100-km wide gradual 660-km discontinuity can be resolved within the shorter period band (2-5 s).

This has been clarified in the revised version of supplementary information file (Line 58-73), and Fig.R2 and R3 have been included in the revised supplementary document as new Supplementary Fig.6 and Supplementary Fig.7.

Fig. R1 Travel time difference for lateral perturbations to the PP bounce point at 410- (a) and 660-km (b) depth, respectively. Red triangles mark the locations of stations. Green and red circles depict the Fresnel's zone for 3.5 s and 7.5 s, respectively. The Fresnel zone is defined by a $\pm T/2$ contour.

Fig. R2 Three models designed to test the resolution of $P_{660}P$ to local gradual 660-km interface. **a** A simple model derived from iasp91 with sharp 410- and 660-km discontinuities. **b** A model similar to **a** but with a 200-km wide gradual 660-km discontinuity (a linearly increase from 660 km to 720 km). **c** A model similar to **b** but with a 100-km wide gradual 660-km discontinuity. Vertical force sources are set on the free surface at 300 km to excite seismic wavefield.

Fig. R3 Seismic waveforms recorded on the free surface calculated by SPEC2FEM2D package (Tromp et al., 2008). **a** and **b** show the synthetic waveforms from Model 1 filtered to 2-5 s and 5-10 s, respectively. **c** and **d** show the synthetic waveforms from Model 2 filtered to two target period bands. **e** and **f** show the synthetic waveforms from Model 3 filtered to two target period bands.

2, line 209-212. High frequency P'P' precursors (PKP PKP) are important for studies of 410 and 660 interfaces (Day and Deuss, 2013, GJI). As for P wave receiver functions, its sensitivity on V_s actually is also important for constraining the nature of 410/660 km. The authors are encouraged to combine reflected P waves (this study) and previous P410s, P660s datasets to provide tighter constraints.

Thanks for your suggestion. We totally agree that the sensitivity of P wave receiver functions (RF) on V_s is also important for constraining the nature of 410- and 660-km discontinuities, and we believe results from RF and ANI should be comparable to some extent. In the future, we would love to combine two datasets to jointly invert/constrain MTZ structure, which is another work and out of the scope of the current study. We add some discussion about previous RF results and comparison with our new results in the revised manuscript (Line 225-232). We also address it below.

As most old RF studies suffer from low resolution, we collected some newest receiver function results within our study region. Though no significant changes in P_{410S} and P_{660S} amplitude are reported by RF studies at the same region, apparently depressed and broadened P_{660S} phase can be identified beneath the North China basin in the newest RF study (Fig. 7e and Fig. S3f from Sun et al. (2020), also attached as Fig. R4 and Fig. R5 below, respectively). Though both 410- and 660-km interface show remarkable depression beneath the North China basin (Fig. R4), the magnitude of depression of the 660-km interface is larger than that of the 410-km interface resulting in a thicker mantle transition zone (Fig. R5). A simple and straightforward explanation for the deeper 660-km interface is a lower temperature, usually induced by subducted slabs. To jointly analyze RFs and reflected waveforms, here we calculated the RFs and reflected waveforms from both iasp91 model and a new derived model with a 60-km wide velocity gradient beneath 660-km depth (Fig. R6). A gradual gradient beneath 660 km will cause a deeper and broadened P_{660S} and very weak P_{660P} , compared with iasp91 model. Therefore, the depressed P_{660S} phase may be alternatively interpreted as a gradual gradient beneath 660-km depth caused by trapped MORB composition, which also well explains the broadening of the P_{660S} phase. It should be noted that the RF results look very smooth, only resolving large-scale features.

In summary, though the depression of 660-interface beneath North China basin inferred by RF may be explained by low temperature or a gradual gradient beneath 660-km depth, the broadened P_{660S} phase favors a gradual gradient beneath 660 km better, plausibly caused by trapped MORB composition in line with our model. It is difficult to identify a gradual gradient zone using the RF alone, however, it will be possible when RF and reflected phases are jointly analyzed. As we do not have the RF waveform data, detailed analysis is not included in the main text.

[Redacted]

Fig. R4 Depth series from stacking of RFs in radius = 1° bins along 39° N latitudinal profile. The background image shows the P-wave velocity anomalies (Chen et al., 2017). (Fig.S3 from the supplementary document for Sun et al. (2020))

[Redacted]

Fig. R5 Corrected MTZ thickness using the wavespeed models of Lu et al. (2019). (Fig.7e from Sun et al. (2020))

Fig. R6 **a** S-wave velocity structure of iasp91 model (Kennett and Engdahl, 1991) and a new derived model with a gradual gradient beneath 660-km depth. **b** The corresponding receiver functions for models shown in **a**, calculated with the CPS package (Herrmann, 2013) with the Gaussian filter parameter set to 1. **c** Synthetic waveforms from the models shown in **a**. The models shown in **a** were extended to 2-D laterally uniform models and calculated the synthetic waveforms with SPECFEM2D (Tromp et al., 2008) with an offset of 100 km. The reflected waveforms were filtered to 5-10 s and normalized.

3, line 150. The distance of 100km is adopted here. Why this particular distance?

Thanks for your comment. As only the NCFs with interstation distances between 0-200km are employed for further common reflection point stacking. Here the middle interstation distance (100 km) was selected as the epicentral distance for waveform simulation, which may be not accurate but acceptable. As shown in Fig. R3(a and b), the reflected waveforms look similar at small offset (< 200 km), this selection may not bias our result significantly.

4, Supplementary figures comparing/contrasting P410P,P660P from ANI for the study region in this study and other regions could be helpful. The paper Poli et al (2012) has already some observations of the reflected P waves from mantle interfaces.

Thanks for your suggestions. Comparing with previous ANI results is a good way to verify our new result. However, there are still some obstacles to directly comparing our result with previous observations. First, the data processing adopted in this study differs from that presented in the paper Poli et al., (2012) in detail. For example, all the NCFs, the offsets varying from ~30 km to 600 km, were stacked in the paper Poli et al. (2012) after eliminating the surface waves (setting waveform within the surface wave window to zero). Thanks to our large dataset, in this study we finally only stack the NCFs with offset less than 200 km to avoid the interference from surface waves. Moreover, Poli et al. (2012) just show one final average trace and 1-D potential structure, here we utilize the stacked traces to investigate the lateral variation of the 410- and 660-km discontinuities. Moreover, the upper mantle structures beneath the two study regions are different. Anyway, more discussion was added about Poli's model in the manuscript (Line 112-113). Our new data and Poli's model both infer a thicker 410-km discontinuity, with respect to the sharp 660-km discontinuity. Besides, more tests were performed to verify the reliability of the recovered reflected body waves (Fig. R6-R8).

Reviewer #2 (Remarks to the Author):

The manuscript presents P wave reflection analysis of mantle transition velocity discontinuities to infer variations in mantle composition related to subducted slab remnants beneath eastern China. The method of obtaining small offset P wave reflections relies on ambient noise interferometry and is still relatively new but has been validated in multiple studies. Generally, the recovery of reflections from 410 and 660 is impressive, as is the large scale spatial variability in the relative amplitudes of the two reflections in multiple frequency bands. The transition from a simple sharp 660 far away from slab remnants to a complicated and/or absent 660 reflection closer to slab remnants appears reliable based on the recovery of 410 reflections and the merit of the processing methods. This spatial transition is the key result for the authors' interpretations. The main conclusion is aligned with many studies indicating that the 660 is a leaky boundary in mantle convection that is largely driven by subduction of slabs with two major compositionally distinct components. It's not surprising but this study offers an impressive new angle of evidence that bears on long-standing problems in understanding mantle dynamics and compositional evolution. The text is generally clear and the figure quality is good. I recommend publication after an opportunity for minor revisions.

I'd note that the data used are not publicly available and hence the study is not reproducible by other researchers in the near future, nor is there a set embargo time window after which the data will become open access. I'm not sure if reproducibility is required by Nature Communications and my evaluation above is only based on the scientific merit of the manuscript.

We are grateful to you for your effort reviewing our paper and positive feedback. We completely agree that making the raw data publicly open to the scientific community is very important, however, as you realized restrictions apply to the raw data and we do not have the right to publish the raw dataset. Frankly speaking, this work is not easily reproducible even

though the raw data is publicly open, because the raw dataset is very large and the calculation of the NCFs is really expensive. To make this study easily reproducible for other researchers, we make all the ambient noise cross-correlation functions (NCFs) publicly available (in the Mendeley Data repository <http://dx.doi.org/10.17632/m9ry8nbfwj.1>). If readers do need the raw data, they should request from the data management centers by providing proof that the applicant is engaged in relevant research, that is how we collected all the raw waveforms data. The data manager will make the final decision. We believe the publication of all the NCFs data can ensure the repeatability of this study. We also wrote a data availability statement to make the circumstances for data availability transparent to readers.

Reviewer #3 (Remarks to the Author):

This is an interesting paper that images mantle transition-zone discontinuities in part of eastern China just west of the subducting Pacific slab and finds evidence for changes in discontinuity properties that the authors argue can be explained as differences in composition related to subduction of oceanic crust. The seismic results are based on ambient noise cross-correlation, a relatively novel approach to studying mantle discontinuities that may have some advantages over more traditional methods. I think the results are potentially important enough to publish in Nature Communications, but I do have some concerns.

We appreciate your effort to review our manuscript and your positive feedback. You give an accurate summary of our work and bring forward constructive questions. We have revised the manuscript carefully following your suggestions. Detailed point-to-point responses to your concerns can be found below.

(1) Does the noise cross-correlation method accurately retrieve the station-to-station Green's function, specifically the P410P and P660P phases of interest? If the distribution of noise sources is not uniform, then biases and artifacts could be introduced and the assumed bounce-point locations for P410P and P660P might not be correct. Thus, it is important test to evaluate how evenly distributed the incoming noise signals are. Here is one way to perform such a test:

Redo Fig. S1 for two independent subsets of the data, based on the azimuth of the station pair used for the cross-correlation: 0–90 degrees and 90–180 degrees (the division is not for 0–360, as 180 deg. folding of the results is performed (lines 223– 224)). If the images of P410P and P660P don't agree, this suggests that the noise may have some dominant back-azimuths and more work will be needed to make sure the results in the paper are not biased by this non-uniformity. Note: Because of the dense station lines at about 120 deg. azimuth, the 90–180 azimuth bin will have many more cross-correlations, so it might be necessary to perform the test with a more uniform station distribution, i.e., discard data from the stations along these lines, so that the images have roughly equal amounts of data.

Thanks very much for your comments and suggestions. We fully understand your concerns about the effects of noise source distributions on the recovery of target phases. For an ideal diffuse field, it has been proved theoretically that the empirical Green's function (EGF) between two stations can be estimated by the cross-correlation of a continuous ambient noise record. According to previous theoretical studies (Nakahara, 2006), here differentiation of the NCFs was calculated to estimate the EGFs. We totally agree that the ambient noise field on the Earth is never an ideal diffuse field. However, at short periods (< 10 s), the reconstruction of body wave phases is not significantly affected by earthquake coda waves and the NCFs are

close to the Green's function expected for the actual Earth response (Boue et al., 2014) . Following your suggestion, here we perform a simple test to evaluate the influence of azimuthally heterogeneous noise source.

As you mentioned the azimuthal distribution of station pairs is not uniform, we divide all the NCFs into two independent datasets (Group1 and Group2, Fig. R7a), making two groups having roughly equal amounts of data. The NCFs from two groups are stacked with respect to inter-station distance similarly as Supplementary Fig. 1 (Fig. R7b, c). The target phases retrieved from two independent datasets looks similar, both aligned well with respect to the theoretical traveltime curves. It seems that the uneven distribution of noise source does not significantly bias the recovery of the target phases. Of course, this test is rough with low azimuthal resolution. More detailed work still needs to be done in the future.

Fig. R7 **a** Azimuthal distribution of all NCFs and the division of two sub-datasets. **b** and **c** Arrangement of two sub-datasets with respect to interstation distance. Black lines represent the theoretical travel-time curves of $P_{410}P$ and $P_{660}P$.

(2) It would be good to plot the theoretical $P_{410}P$ and $P_{660}P$ travel-time curves on top of Fig. S1 to make sure they align exactly with the observations.

Thanks for your good suggestion. The theoretical travel-time curves of $P_{410}P$ and $P_{660}P$ are overlapped on the replotted Supplementary Fig. 1 (Fig. R8). As the reflected body waves retrieved from ambient noise interferometry are fat wavelets rather than pulses which are usually expected for receiver functions, here we mainly focus on the slope of target wavelets. The reflected wavelets exhibit good alignment with respect to the theoretical travel-time curves, which further validate the retrieved target phases.

Fig. R8 All the NCFs filtered into 5-10 sec (a) and 2-5 sec (b) period bands and arranged by inter-station distance. The NCFs were stacked within a series of overlapped distance bins and normalized by the number of stacked NCFs, where the width of bins is set to be 19 km. Black lines represent the theoretical travel-time curves of $P_{410}P$ and $P_{660}P$. The number of stacked NCFs within each bin is shown in c.

(3) What causes the reduction in amplitude in $P_{410}P$ and $P_{660}P$ near zero offset? I would not have expected the reflection coefficient to change very much with distance at the relatively steep incidence angles between 0 and 200 km offset. The fold drops near zero offset, but this should increase the noise, not dampen the signal, right?

Thanks for your comment. You are exactly right that the reflection coefficient between 0 and 200 km offset should not change very much, which are also validated by the synthetic

waveforms in Fig. R3(a and b). The observed reduction in amplitude of reflected phases is nonphysical, which is caused by the reduction in the amount of stacked data at small offset (Fig. R8c). The old Fig. S1a and b previously exhibit stacking energy without normalization by the amount of stacked data. We replotted Supplementary Fig. 1 and Supplementary Fig. 2 after normalizing each trace by the number of stacked NCFs within each distance bin, and the reduction in amplitude is no longer prominent. Of course, the reflection phases are still weak at very small offset (< 30 km) where these reflection phases can be hardly retrieved due to the very small dataset.

After normalization, the reflected body waves at a small offset still exhibit pretty high signal-to-noise ratio and are not significantly dampened. As we mentioned in the manuscript that absolute amplitude information was lost during time-domain normalization and we solely focus on the relative amplitude information (Line 255-258). Replotting Supplementary Fig. 1 and Supplementary Fig. 2 has negligible influence on the discussion in the main text where only the relative amplitude features are emphasized.

(4) Given the observed amplitude reduction near zero offset, it makes me nervous that the 200 km cutoff used for the common-reflection-point (CRP) stacking is at a range where the amplitudes are changing. To make sure there is nothing systematically varying with station location that might be biasing the CRP stacks, it would be good to see Fig. S1 repeated using just the data used for the profile shown in Fig. 3, and then plotted separately using just the data for the reflection points in the profile east and west, respectively, of 116.5 longitude (which approximately separates the main changes that are observed). The difference in the P410P and P660P amplitude and character east and west of this point in the profiles is perhaps the main observational result in the paper, so it's important to verify its robustness and make sure it does not have some other possible origin unrelated to the actual mantle discontinuity properties.

Thanks for your comment and suggestions. As we have addressed above that the observed amplitude reduction is nonphysical, so the 200-km cutoff for the CRP stacking is appropriate. Anyway, you suggest a good way to validate the difference in the relative amplitude of P_{410P} and P_{660P} . Following your suggestion, we define two independent regions (Region A and Region B) to repeat Supplementary Fig.1 (Fig. R9). It is not easy to recover clear coherent P_{410P} and P_{660P} phases with respect to inter-station distance because of the small datasets. As expected, coherent and clear P_{410P} can be observed in both regions. However, clear and energetic P_{660P} phases can only be observed in region A. This test further validates the lateral variations in relative amplitude retrieved from ANI.

This has been clarified in the revised version of supplementary information file (Line 83-88), and Fig.R9 has been included in the revised supplementary document as new Supplementary Fig. 10.

Fig. R9 a Station map and the definition of two independent regions (A and B). Green dots denote the reflection points of NCFs. b and c display all the NCFs reflected within regions A and B, respectively, arranged with respect to interstation distance. The NCFs were stacked within a series of overlapped distance bins, and the width of the bins is set to 37 km. Black lines represent the theoretical travel-time curves of $P_{410}P$ and $P_{660}P$.

(5) The text near line 102 should reference and discuss the Poli et al. 2012 paper, which found a thicker 410- than 660-discontinuity based on P410P and P660P ambient noise cross-correlations, a result more directly relevant to this paper than the other references cited.

Thanks for your suggestion. The manuscript has been revised accordingly (Line 112-113).

(6) There have been several seismic receiver function (RF) studies of mantle discontinuities in the same region (e.g., Chen and Ai, JGR, 2009; Zhang et al., Tectonophysics, 2016; just what I found quickly---there are likely more). Near the end of the paper (~line 210), the authors mention that their method differs from RF studies because it is sensitive to P velocity changes rather than S velocity changes. Nonetheless, one would expect P and S velocity changes across the 410 and 660 to be correlated to some extent and to occur at similar depths. Thus, to put the current paper into better context with previous work, it would be good to include some mention of what the RF studies have found. That is, do they see changes in 410 or 660 amplitude, topography, and/or sharpness that agree or disagree with the topside P reflection data? If there is agreement, this would help support the author's model. If there is disagreement, some discussion and perhaps arguments as to why the new results are better would help the reader in sorting out what is going on.

Thanks for your suggestion. We believe that the results from RF and ANI should be comparable to some extent and we add some discussion about previous RF results and comparison with our new results in the revised manuscript (Line 225-232).

As most old RF studies suffer from low resolution, we collected some newest receiver function results within our study region. Though no significant changes in P_{410S} and P_{660S} amplitude are reported by RF studies at the same region, apparently depressed and broadened P_{660S} phase can be identified beneath the North China basin in the newest RF study (Fig. 7e and Fig. S3f from Sun et al. (2020), also attached as Fig. R4 and Fig. R5, respectively). Though both 410- and 660-km interface show remarkable depression beneath the North China basin (Fig. R4), the magnitude of depression of the 660-km interface is larger than that of the 410-km interface resulting in a thicker mantle transition zone (Fig. R5). A simple and straightforward explanation for the deeper 660-km interface is a lower temperature, usually induced by subducted slabs. To jointly analyze RFs and reflected waveforms, here we calculated the RFs and reflected waveforms from both iasp91 model and a new derived model with a 60-km wide velocity gradient beneath 660-km depth (Fig. R6). A gradual gradient beneath 660 km will cause a deeper and broadened P_{660S} and very weak P_{660P} , compared with iasp91 model. Therefore, the depressed P_{660S} phase may be alternatively interpreted as a gradual gradient beneath 660-km depth caused by trapped MORB composition, which also well explains the broadening of the P_{660S} phase. It should be noted that the RF results look very smooth, only resolving large-scale features.

In summary, though the depression of 660-interface beneath North China basin inferred by RF may be explained by low temperature or a gradual gradient beneath 660-km depth, the broadened P_{660S} phase favors a gradual gradient beneath 660 km better, plausibly caused by trapped MORB composition in line with our model. It is difficult to identify a gradual gradient zone using the RF alone, however, it will be possible when RF and reflected phases are jointly analyzed. As we do not have the RF waveform data, detailed analysis is not included in the main text.

Reference:

- Boue, P., Poli, P., Campillo, M., Roux, P., 2014. Reverberations, coda waves and ambient noise: Correlations at the global scale and retrieval of the deep phases. *Earth Planet Sc Lett* 391, 137-145.
- Herrmann, R.B., 2013. Computer Programs in Seismology: An Evolving Tool for Instruction and Research. *Seismological Research Letters* 84, 1081-1088.
- Kennett, B.L.N., Engdahl, E.R., 1991. Traveltimes for Global Earthquake Location and Phase Identification. *Geophys J Int* 105, 429-465.
- Lu, C., Grand, S.P., Lai, H.Y., Garnero, E.J., 2019. TX2019slab: A New P and S Tomography Model Incorporating Subducting Slabs. *J Geophys Res-Sol Ea* 124, 11549-11567.
- Nakahara, H., 2006. A systematic study of theoretical relations between spatial correlation and Green's function in one-, two- and three-dimensional random scalar wavefields. *Geophys J Int* 167, 1097-1105.
- Sun, M.C., Gao, S.S., Liu, K.H., Fu, X.F., 2020. Upper mantle and mantle transition zone thermal and water content anomalies beneath NE Asia: Constraints from receiver function imaging of the 410 and 660 km discontinuities. *Earth Planet Sc Lett* 531.
- Tromp, J., Komatitsch, D., Liu, Q.Y., 2008. Spectral-element and adjoint methods in seismology. *Commun Comput Phys* 3, 1-32.

REVIEWERS' COMMENTS

Reviewer #1 (Remarks to the Author):

The authors have made sufficient revisions that addressed my concerns, including Fresnel's zone, lateral variation, etc. Thus, I believe that the paper is convincing about the findings of oceanic crust above mantle discontinuities. I am happy to suggest that this important paper on nature of mantle transition zone is very good shape and qualified for publication in Nature Communications.

Reviewer #3 (Remarks to the Author):

The authors have adequately responded to my comments and concerns, so I am happy to see the paper published.

I waive anonymity.

Peter Shearer

Reviewer #1 (Remarks to the Author):

The authors have made sufficient revisions that addressed my concerns, including Fresnel's zone, lateral variation, etc. Thus, I believe that the paper is convincing about the findings of oceanic crust above mantle discontinuities. I am happy to suggest that this important paper on nature of mantle transition zone is very good shape and qualified for publication in Nature Communications.

Reviewer #3 (Remarks to the Author):

The authors have adequately responded to my comments and concerns, so I am happy to see the paper published.

I waive anonymity.

Peter Shearer

We appreciate Prof. Peter Shearer and another anonymous reviewer for their positive comments on our manuscript.